# C-Reactive Protein Pretreatment-Level Evaluation with Histopathological Correlation for Chondrosarcoma Prognosis Assessment—A 15-Year Retrospective Single-Center Study

**DOI:** 10.3390/diagnostics14131428

**Published:** 2024-07-04

**Authors:** Sarah Consalvo, Florian Hinterwimmer, Maximilian Stephan, Sebastian Breden, Ulrich Lenze, Jan Peeken, Rüdiger von Eisenhart-Rothe, Carolin Knebel

**Affiliations:** 1Department of Orthopaedics and Sport Orthopaedics, Klinikum Rechts der Isar, Technical University of Munich, 81675 Munich, Germany; maximilan.stephan@muenchen-klinik.de (M.S.); sebastian.breden@mri.tum.de (S.B.); ulrich.lenze@mri.tum.de (U.L.); eisenhart@tum.de (R.v.E.-R.); carolin.knebel@mri.tum.de (C.K.); 2Institute for AI and Informatics in Medicine, Technical University of Munich, 81675 Munich, Germany; florian.hinterwimmer@tum.de; 3Department of Radiooncology and Radiotherapy, Klinikum Rechts der Isar, Technical University of Munich, 81675 Munich, Germany; jan.peeken@mri.tum.de

**Keywords:** chondrosarcoma, CRP, prognosis, microenvironment

## Abstract

Background: An aberrant cellular microenvironment characterized by pathological cells or inflammation represents an added risk factor across various cancer types. While the significance of chronic inflammation in the development of most diffuse tumors has been extensively studied, an exception to this analysis exists in the context of chondrosarcomas. Chondrosarcomas account for 20–30% of all bone sarcomas, with an estimated global incidence of 1 in 100,000. The average age at diagnosis is 50, and over 70% of patients are over 40. This retrospective study aimed to examine the role of C-reactive protein (CRP) as a prognostic factor in relation to the histopathological findings in chondrosarcoma. Methods: In this retrospective study, 70 patients diagnosed with chondrosarcoma and treated between 2004 and 2019 were included. Preoperative CRP levels were measured in mg/dL, with non-pathological values defined as below 0.5 mg/dL. Disease-free survival time was calculated from the initial diagnosis to events such as local recurrence or metastasis. Follow-up status was categorized as death from disease, no evidence of disease, or alive with disease. Patients were excluded if they had insufficient laboratory values, missing follow-up information, or incomplete histopathological reports. Results: The calculated risk estimation of a reduced follow-up time was 2.25 timed higher in the patients with a CRP level >0.5 mg/dL (HR 2.25 and 95% CI 1.13–4.45) and 3 times higher in patients with a tumor size > pT2 (HR 3 and 95% CI 1.59–5.92). We can easily confirm that risk factors for reduced prognosis lie in chondrosarcoma high grading, preoperative pathological CRP- level, and a size > 8 cm. Conclusions: A pretreatment CRP value greater than 0.5 mg/dL can be considered a sensitive prognostic and risk factor for distant metastasis for chondrosarcoma patients.

## 1. Introduction

Chondrosarcomas account for 20–30% of all bone sarcomas, with an estimated annual incidence of 1 in 200,000 in the United States [1,2]. Emerging literature indicates an increasing trend in chondrosarcoma rates, establishing it as the most common primary bone malignancy in certain countries, primarily due to incidental diagnoses of chondrosarcoma-associated lesions [3,4]. Typically diagnosed at the age of 51, chondrosarcoma predominantly affects individuals over 40, constituting more than 70% of cases. However, a noteworthy exception is the mesenchymal chondrosarcoma subtype, which manifests at a significantly younger age, peaking in incidence during the second and third decades of life. Gender distribution in chondrosarcoma displays a mild male predilection, although variations exist among different subtypes. The incidence of chondrosarcoma cases among females has been observed to increase by over 50%. This trend is theorized to be linked to the rising usage of oral contraceptives and menopausal hormonal therapy. A study suggests that an estrogen-signaling pathway may contribute to the proliferation of malignant chondrocytes [1]. The conventional primary chondrosarcoma, representing 85% of all cases, is the most common variant. Less common subtypes include secondary chondrosarcomas originating from benign precursors, as well as dedifferentiated, periosteal, mesenchymal, and clear cell variants [2,5]. 

C-reactive protein serves as a widely utilized systemic biomarker for detecting both acute and chronic inflammation. Over the last decade, there has been a renewed emphasis on the clinical utility of serum CRP, extending beyond inflammation to encompass the prediction and diagnosis of various conditions, notably cardiovascular diseases and malignancies. Elevated serum CRP levels have been observed in patients with numerous malignancies, highlighting a close association between inflammation and cancer [6,7,8,9,10]. The term “tumor CRP” is commonly used but until now unexplained.

Preoperative CRP levels correlate with the progression and pathological stages of malignancies. Elevated CRP emerges as a significant predictor of lower survival rates across various cancers, such as esophageal, colorectal, hepatocellular, pancreatic, urinary bladder, renal, ovarian, and cervical cancer, following surgical resection [7,11,12]. The acute phase reaction serves the dual purpose of inducing local tissue damage through a localized response while concurrently preventing the spread of damage beyond a certain threshold. These reactions act as pertinent diagnostic markers, reflecting the magnitude of the body’s response. Acute phase proteins are generated within a 24 h timeframe, resulting in an approximately 25% elevation in their blood concentration during this period. Consequently, protein concentrations can surge up to 1000-fold [10,13]. Given the established connection between inflammation and cancer, it becomes imperative to scrutinize the prognostic relevance of various inflammatory factors. Taking into account the recent insight into the existence of multiple isoforms of CRP, each with unique biological functions, a comprehensive model is proposed. This model elucidates how CRP serves as a mediator of host defense responses in cancer [13]. The simplicity, affordability, and widespread availability of serum CRP measurements make it a valuable tool in daily clinical practice. It can function as an additional prognostic indicator for survival and post-treatment monitoring in cancer patients. 

The prognostic significance of C-reactive protein, however, remains in chondrosarcomas unexplored. The objective of this retrospective study is to assess the potential utility of CRP in gauging a patient’s risk of diminished life expectancy at the time of diagnosis and correlation with histopathological grading. 

## 2. Materials and Methods

This retrospective study included 70 patients treated at Klinikum rechts der Isar between 2004 and 2019 with chondrosarcoma diagnoses. Diagnoses for all patients were validated through the histopathological department. The local institutional review and ethics board (Klinikum rechts der Isar, Technical University of Munich) approved this study (No. 48/20S, 17 February 2020). Exclusion criteria were insufficient laboratory values, missing follow-up or histopathology information, and Rx or R1/R2 resection status. Also, all atypical cartilaginous tumors (ACT) on the base of the benign nature of this lesion were excluded. All types of chondrosarcoma were included, in detail: mesenchymal, mixoid, clear cell, and classic chondrosarcomas. In total, 3 patients were excluded.

Histology was obtained through CT-scan-guided or open biopsy performed in our institution and discussed at our interdisciplinary sarcoma board following the WHO guidelines.

Blood samples were collected during pretreatment between 1 to 7 days before biopsy or first surgical treatment. CRP levels were reported in milligrams per deciliter (mg/dL) with a non-pathological value below 0.5 mg/dL. The measurement was performed using the Cobas^®^ 8000 modular analyzer C702 (Fa. Roche, Munich, Germany). 

Aftercare for all patients was performed every 12 weeks for the first 2 years, twice a year from the 3rd to the 5th year, and annually thereafter in accordance with German guidelines. Time between the initial diagnosis and local recurrence or distant metastasis was defined as disease-free survival (DFS). All metastasis locations were included. Time between the initial diagnosis and the last follow-up or the last (unscheduled) presentation of the patient for aftercare in our clinic was considered as “follow-up time”. The follow-up status was grouped as death of disease (DOD), no evidence of disease (NED), and alive with disease (AWD) for patients who are currently alive but had a diagnostically confirmed distant metastasis or a proven local recurrence.

### Statistical Analysis

Data were processed and analyzed using StatPlus Pro 2020 (AnalystSoft). Statistical reporting followed STROBE guidelines. A national standard of 0.5 mg/dL was used to distinguish between “increased” and “decreased” CRP levels. The correlation between CRP values and overall survival (OS) was evaluated using a Pearson correlation model. Guide values for the correlation coefficient (CC) were interpreted as follows: CC = 0.3 indicates no linear correlation, CC = 0.3–0.5 indicates a weak positive linear correlation, and CC > 0.5 indicates a positive linear correlation.

Statistical significance was set at *p* < 0.01. Sensitivity and specificity were calculated, along with the 95% confidence interval. Survival curves were generated using Kaplan–Meier analysis. Cox regression analysis was performed to estimate the association between CRP levels, prognosis, T status, and grading.

## 3. Results

We treated 73 patients at Klinikum rechts der Isar between 2004 and 2019 with a confirmed diagnosis of chondrosarcoma. From this cohort, we excluded three patients due to the lack of precise information in the histopathology report. The clinical characteristics of the patients are summarized in Table 1. Patient mean age was 56 years with a minimum age of 26 years and a maximum age of 87 years. Majority of the cohort was male (43 patients vs. 27 female patients). The mean female age was 56.5 years (26–82 years), whereas that of males was 56 years (29–87); 50% had lower extremity localization (40% femur, 5.7% tibia, and 4.2% foot), 24.2% pelvis localization, 20% upper extremity localization, and 5.7% thorax cage localization.

The follow-up status was defined as DOD in 37 (52.8%), NED in 32 (45.7%), and AWD in 1 (1.4%) patient. In all patients, a surgical R0 therapy was performed. R0 resection was confirmed by the pathology department (n = 70).

The median CRP was 0.5 mg/dL (0.1–13.2 mg/dL).

In the DOD patients, the median CRP level was 0.7 mg/dL (0.1–13.2 mg/dL). Only one patient remained alive with distant metastasis and local recurrence; his preoperative CRP level was 0.5 mg/dL. The median CRP value of the NED group was 0.3 mg/dL (0.1–6 mg/dL) (Figure 1). It was observed that patients in our cohort who ultimately died from chondrosarcoma had, on average, higher preoperative CRP values.

In the elevated CRP group (>0.5 mg/dL), 28.5% of the patients had died, while 10% were alive without any disease manifestation. In the low-CRP group (<0.5 mg/dL), 24% had died, 1.4% were alive with metastases or local recurrence, and 35.7% were alive without any disease manifestation (Table 2).

In evaluating the tumor size in the group with an elevated CRP, the majority had a tumor size > 8 cm (pT2, according to the 5th Edition 2020 WHO [14]) while the group with low CRP levels was mostly represented by chondrosarcomas with <8 cm size (pT1) (Table 2 and Figure 2).

Patients with G1 chondrosarcoma had a median CRP level of 1.5 mg/dL (0.1–6 mg/dL). Two patients with G4 grading were excluded in the mentioned below box plot table due to the non-statistical value. The median CRP value of the G2 group was 0.5 mg/dL (0.1–2.3 mg/dL), and in the G3 group, the median CRP was 2.7 mg/dL (0.1–13.2 mg/dL) (Figure 3). It can be noticed that the CRP levels between G1 and G2 showed not much difference, but the G3 group had higher preoperative CRP values on average.

If we summarize the data of our cohort until this point, we can say that patients who had an elevated preoperative CRP level had a bigger tumor (>pT2), a higher grading (>G3), and a poorer prognosis.

We calculated, therefore, the statistical correlation between the CRP level and the T status, the follow-up, and grading.

The correlation coefficient between CRP and follow-up was 0.33 (*p*-value 0.004), between CRP and T status 0.24 (*p*-value 0.04), and between CRP and grading 0.15 (*p*-value 0.20).

There is a statistically significant direct proportion between CRP and follow-up.

We could confirm the data with a Cox regression. Here, we found a significant correlation with all of the three factors (Table 3).

### Survival Curves

The median follow-up time and DFS time were, respectively, 2.5 years (0–12.9 years) and 3 years (0–11.8 years). Patients with a preoperative CRP value over 0.5 mg/dL had a median follow-up time of 1.2 years (0–9.9 years). On the other hand, patients with a CRP value below 0.5 mg/dL showed a median follow-up time of 4.3 years (0–13 years) (Figure 4).

A 5-year survival rate of 94% was calculated for the cohort with the CRP levels < 0.5 mg/dL and 85% for the cohort with the CRP levels < 0.5 mg /dL.

The calculated risk estimation of a reduced overall survival rate time was 2.25 times higher in the patients with a CRP level > 0.5 mg/dL as well as the risk of recurrence of an event in terms of local recurrence or distant metastasis (HR 2.28 and 95% CI 1.15–5.53).

On the other hand, the calculated risk estimation of a reduced overall survival rate time for patients with a tumor size > pT2 was three times higher (HR 3 and 95% CI 1.59–5.92). The risk of recurrence of an event in terms of local recurrence or distant metastasis was also similar (HR 3.1 and 95% CI 1.63–6) (Figure 5).

## 4. Discussion

As tumors grow and spread, they interfere with normal tissue function and trigger the body’s acute phase inflammatory response. C-reactive protein (CRP) is a crucial element of this response and is often used as a minimally invasive marker to detect inflammation. However, its specific role in assessing cancer progression or remission remains unclear [9,13,15]. In 2011, in a Copenhagen General Population Study, the CRP levels of a cohort of 63,500 healthy patients were analyzed. Patients with solid tumors (breast, lung, or prostate cancer) and elevated CRP levels at the time of diagnosis were associated with poor prognosis. On the other hand, healthy patients with elevated CRP levels were associated with increased future risk of cancer. Patients with cancer and elevated CRP levels had an 80% greater risk of early death [6]. In their study, the cut-off limit of “elevated” CRP was >10 mg/dL. In our study, we set the cut-off at 0.5 mg/dL in line with the national guidelines.

Our research aims to assess the predictive significance of pretreatment serum CRP analysis concerning prognosis in chondrosarcoma patients. By scrutinizing the statistical relevance of CRP values in correlation with prognosis, grading, and size, we cautiously discern its involvement in fostering a cancer-friendly microenvironment, indicated by local recurrence and distant metastasis.

Due to poor data, which is due to the rarity of these tumor entities, bone and soft tissue sarcomas remain a big question mark on this theme.

The relevance of CRP levels in Ewing sarcoma patients was already demonstrate in our 2022 study [8]. CRP level > 0.5 mg/dL revealed an 8.3 HR (95% CI 3–22.7). Optimal breakpoint analysis of CRP as a prognostic factor of OS revealed an AUC of 0.812. These data were also confirmed by the study of Funovics et al. with significantly correlated disease-specific survival [16]. The study involved 79 patients (37 females, 42 males; average age 18 years) undergoing osteosarcoma resection, with an average follow-up of 46 months. Higher preoperative CRP levels were significantly associated with lower survival rates. Patients who died had an average CRP level of 1.09 mg/dL, compared to 0.32 mg/dL in survivors. A significant negative correlation was found between CRP levels and both survival and histological subtype but not with histological response and tumor size. Patients with CRP levels over 1 mg/dL had a five-year survival rate of 36.7%, compared to 73.8% in those with normal CRP levels. Unlike our study, the cut-off level was set 0.5 mg/dL higher.

In a meta-analysis of five studies with a total of 816 patients with bone neoplasm, higher levels of preoperative CRP levels were associated with reduced OS [17]. For more detail, a ratio between CRP and Albumin concentration was analyzed. Higher CRP/Alb ratio values were significantly associated with poorer overall survival. The CRP/Alb ratio had significantly higher AUC values compared to the also evaluated Glasgow Prognostic Scale (GPS) (*p* = 0.003), indicating superior discriminatory ability.

In alignment with our investigation, Aggerholm-Pendersen et al. examined 172 patients with bone sarcoma, including 63 chondrosarcomas and 109 cases of either Ewing’s sarcomas and osteosarcomas. They found that elevated CRP levels were associated with higher overall mortality in both cohorts. However, they grouped osteosarcomas and Ewing’s sarcomas without distinguishing between them [18]. Elevated levels of pretreatment CRP and an elevated score in the Glasgow prognostic score (GPS), as well as low pretreatment hemoglobin and sodium concentrations, were associated with increased overall mortality and disease-specific mortality [19].

Similarly, a study in 2013 revealed statistically poorer disease-free survival (DFS) in patients with elevated CRP values, encompassing Ewing’s sarcoma and chondrosarcoma, without specifying cut-off values or entity distinctions [20]. Patients with metastatic spread at diagnosis were excluded from that study. The calculated 5-year survival rates for elevated and reduced CRP levels were, respectively, 57% and 79%. In our study, we observed 5-year survival rates of 85% and 94% [20]. Nonetheless, direct comparison is limited due to the inclusion of Ewing sarcoma in the study by Nakamura et al.

Nemecek E. et al. were the only other study group we found in our literature research that analyzed the CRP levels in only chondrosarcoma patients. They analyzed a total of 33 patients in a time range of 38 years (from 1977 to 2015) and demonstrated decreased overall survival in patients with elevated CRP levels (HR 1.31; 95%CI 1.02–1.57; *p* = 0.031) [21].

Further investigations into the inflammatory system’s role in carcinogenesis have utilized ratios. Recent studies examined the CRP–albumin ratio (CAR) [22], showing that higher CAR levels were associated with poor prognosis in Ewing’s sarcoma. Additionally, Biswas et al. found that a high white blood cell (WBC) count predicted inferior event-free survival (EFS) and local control rates in extraosseous Ewing’s sarcomas [23].

By observing the curves in Figure 4 and Figure 5, we can easily confirm that the most elevated risk for a reduced follow-up time lies in chondrosarcomas with high grading, a preoperative pathological CRP- level, and a size >8 cm.

In light of our findings, we affirm the role of pretreatment CRP values in predicting prognosis, local recurrence, and distant metastasis in chondrosarcoma patients. Nonetheless, this study has limitations, primarily stemming from the rarity of the cancer subtype and the small size of the analyzed cohort. Though a confirmed R0 resection may mitigate bias in recurrence and metastasis, the loss of numerous patients during the 15-year follow-up period precluded a complete analysis of outcomes. If we summarize the data of our cohort until this point, we can statistically say that patients who had an elevated preoperative CRP level had a bigger tumor (>pT2), a higher grading (>G3), and a reduced life prognosis.

## 5. Conclusions

It is evident that assessing preoperative CRP levels in patients with chondrosarcoma holds significant prognostic value. Consequently, adjusting the follow-up protocol for patients with preoperative CRP levels exceeding 0.5 mg/dL may be warranted for this high-risk cohort. Nevertheless, additional research and centralized data collection and analysis are necessary to determine the most accurate cut-off values for pretreatment CRP levels in predicting poor survival or distant metastasis.

## Figures and Tables

**Figure 1 diagnostics-14-01428-f001:**
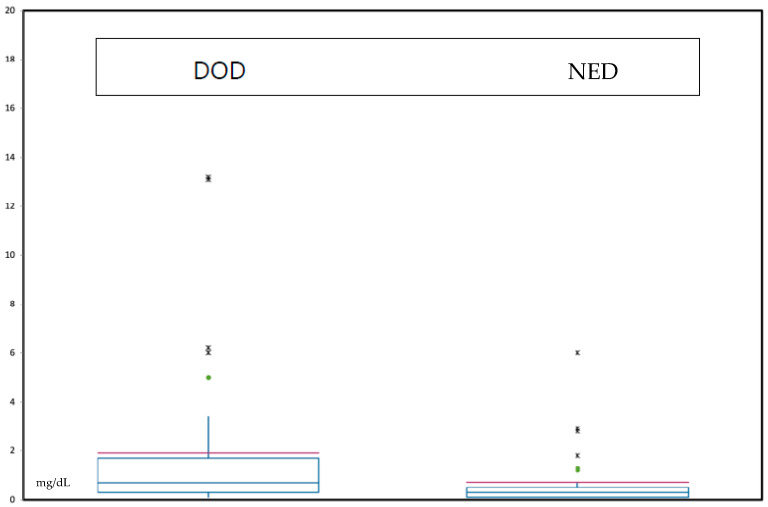
Box plot of the median CRP values (red line) depending on the follow-up status of the patients: no evidence of disease (NED) and death of disease (DOD). For the alive with disease (AWD) status, no diagram was created because only one patient was involved.

**Figure 2 diagnostics-14-01428-f002:**
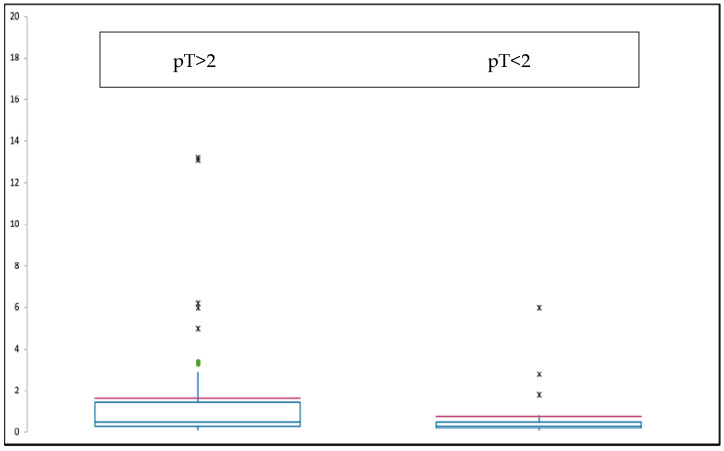
Box plot of the median CRP values (red line) depending on the pT status (tumor size) of the patients, divided in 2 groups: tumor size under pT2 (<8 cm) and over pT2 (<8 cm).

**Figure 3 diagnostics-14-01428-f003:**
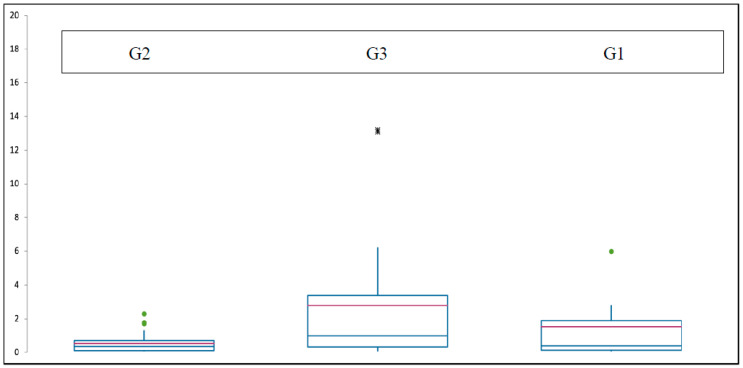
Box plot of the median CRP values (red line) depending on the grading.

**Figure 4 diagnostics-14-01428-f004:**
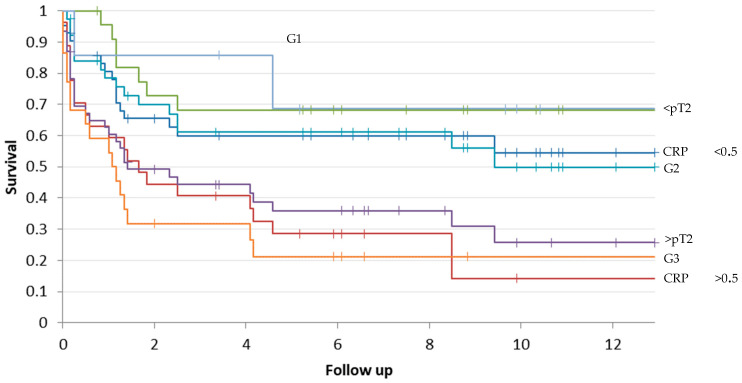
Kaplan–Meier curve of survival rate (follow-up) depending on CRP levels, size of the tumor, and grading: CRP < 0.5 mg/dL (blue curve), CRP > 0.5 mg/dL (red curve), size < pT2 (green curve), size > pT2 (purple curve), G1 (gray curve), G2 (light blue), and G3 (orange curve). The follow-up time is evaluated in years (*p*-value 0.00008).

**Figure 5 diagnostics-14-01428-f005:**
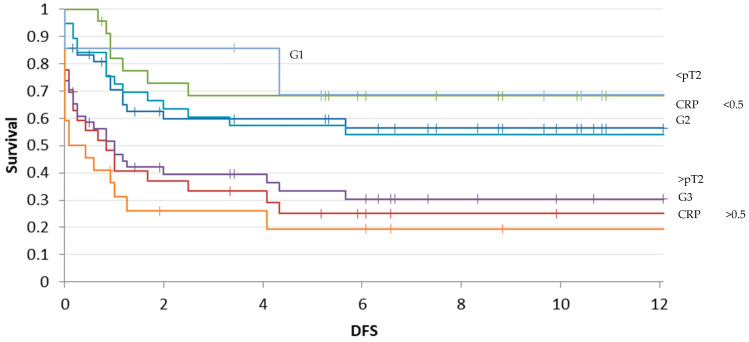
Kaplan–Meier survival curve for disease-free survival depending on CRP levels, size of the tumor, and grading: CRP < 0.5 mg/dL (blue curve), CRP > 0.5 mg/dL (red curve), size < pT2 (green curve), size > pT2 (purple curve), G1 (gray curve), G2 (light blue), and G3 (orange curve). The follow-up time is evaluated in years (*p*-value 0.00002).

**Table 1 diagnostics-14-01428-t001:** Characteristics of included patients with preoperative CRP levels.

	*n*	Percentage	Mean	Min	Max
Age	70	-	56 years	26 years	87 years
**Sex**					
Male	43	61.40%	-	-	-
Female	27	38.80%	-	-	-
**Site**					
Femur	28	40%	-	-	-
Tibia	4	5.70%	-	-	-
Hip	17	24.20%	-	-	-
Scapula	8	11.40%	-	-	-
Clavicula	1	1.40%	-	-	-
Humerus	5	7.10%	-	-	-
Thorax	4	5.70%	-	-	-
Foot	3	4.20%	-	-	-
**Follow-up**					
DOD	37	52.80%	-	-	-
NED	32	45.70%	-	-	-
AWD	1	1.40%	-	-	-
**Metastasis** **at diagnosis**	28	40%	-	-	-
pulmonary	19	27%	-	-	-
Lymph nodes	4	5.70%	-	-	-
multifocal	3	4.20%	-	-	-
other	2	2.80%	-	-	-
**Grading**					
G1	7	10%	-	-	-
G2	38	54.20%	-	-	-
G3	22	31.40%	-	-	-
G4	2	2.80%	-	-	-
no grading	1	1.40%	-	-	-
**T Status**			-	-	-
pT1	21	30%	-	-	-
pT1b	2	2.80%	-	-	-
pT2	34	48.50%	-	-	-
pT2a	2	2.80%	-	-	-
pT2b	10	14.20%	-	-	-
pT3	1	1.40%	-	-	-
**Follow-up**					
DFS	70	-	1.3 years	0 years	12 years
Follow-up time	70	-	2.3 years	0 years	13 years
**CRP**					
<0.5 mg/dL	43	61.4%	0.3 mg/dL	0.1 mg/dL	0.5 mg/dL
>0.5 mg/dL	27	38.50%	1.7 mg/dL	0.7 mg/dL	13.2 mg/dL

**Table 2 diagnostics-14-01428-t002:** Characteristics of included patients with a preoperative CRP level below or under 0.5 mg/dL. T Status was calculated with the WHO guidelines 2020.

Spalte1	CRP < 0.5 mg/dL	CRP > 0.5 mg/dL
**Follow-up**		
DOD	17 (24.2%)	20 (28.5%)
NED	25 (35.7%)	7 (10%)
AWD	1 (1.4%)	-
**T Status**		
pT1	16 (22.8%)	5 (7.1%)
pT1b	2 (2.8%)	-
pT2	18 (25.7%)	6 (22.8%)
pT2a	-	2 (2.8%)
pT2b	7 (10%)	3 (4.2%)
pT3	-	1 (1.4%)

**Table 3 diagnostics-14-01428-t003:** Cox regression equations with follow-up and CRP levels, T status, and grading.

	Beta	Wald	*p*-Value	Exp(B)	Lower	Upper
CRP	0.6749	4.06006	0.04391	1.96296	1.01858	3.78677
T	1.2246	7.29320	0.00692	3.40297	1.39911	8.27679
Grading	0.6713	7.62120	0.00577	1.95686	1.21495	3.15180

## Data Availability

The data that support the findings of this study are available on request from the corresponding author (S. Consalvo).

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
