# Peer review of "C-Reactive Protein Pretreatment-Level Evaluation with Histopathological Correlation for Chondrosarcoma Prognosis Assessment—A 15-Year Retrospective Single-Center Study"

_diagnostics, 2024, doi:10.3390/diagnostics14131428_

Round 1

Reviewer 1 Report

Comments and Suggestions for Authors

The manuscript by Consalvo et al., entitled “  CReactive Protein Pretreatment-Level Evaluation with histopathological Correlation for Chondrosarcoma Prognosis Assessment A 15-Year Retrospective Single-Centre Study“  provides information about the prognostic significance of C-reactive protein (CRP) however remains in Chondrosarcomas unexplored. The objective of this retrospective study is to assess the potential utility of CRP in gauging a patient's risk of diminished life expectancy at the time of diagnosis and correlation with histopathological grading.

The manuscript is well written, and the data are well presented. This study provides interesting information for the scientific community. Accordingly, the manuscript can be published in the present form.

Comments on the Quality of English Language

Is acceptable

Author Response

Thank you very much for your positive feedback on our manuscript. We are delighted to hear that you found our work satisfactory and had no corrections or suggestions for improvement.

Your approval is greatly appreciated, and we are grateful for the time and effort you have dedicated to reviewing our submission.

Once again, thank you for your kind words and support. We look forward to the next steps in the publication process.

Best regards,

S.Consalvo

Reviewer 2 Report

Comments and Suggestions for Authors

The manuscript by Consalvo et al aims to correlate values of c-reactive protein, a fairly routine test, with oncologic outcomes in chondrosarcoma. This is a worthwhile project in light of burgeoning financial constraints placed on the health system. Having a relatively inexpensive and easily measured biomarker as a predictor of outcomes in a rare disease would help change practice. 

There are several edits that need to be done for it to become acceptable for publication:

- please clarify why CRP is a routinely measured test in these patients. In general, CRP is inconsistently measured in suspected cases of sarcomas. Why were CRPs done for all of these cases?

- in table 1: please explain grade 4; which system is used?

- please include the subtypes of chondrosarcomas included in the study. Are there differences in the subtypes of chondrosarcomas and the levels of CRP?

- figure 1 is incomplete: please label axes and the box plot ("NED" seems to be missing).

- figure 2 is also incomplete: please label axes and the box plot appropriately

- figure 3: in the table there were 4 grades...but this table only contains 3?

- figure 4: the KM curves are too busy and confuse the picture - would consider breaking these into different graphs

- figure 5: comments same as figure 4

- in lines 194-199: the authors are concentrating on the relevance of CRP...but shift part of the results to the effect of tumor size on prognosis. Perhaps consider rewording this section

- lines 200-202: I think this belongs in the discussion section.

Comments on the Quality of English Language

Edits are needed for:

- abbreviation

- spelling

- grammar

Author Response

The answers are listed in the same sequence as the questions in the reviewers document:

- Question 1: clarify why CRP is routinely measured:

In our clinic before any kind of intervention or operation (from CT biopsy to any kind of operation) we do a routine blood checkup that involves measuring CRP levels to threat and  not misdiagnose any kind of preoperative infection (like Pneumoniae, infectious skin diseases or urinal tract infection). We try to treat and exclude any infection before every intervention/operation.

-  Question 2: Please explain G4:

Our pathologist use in very undifferentiated Sarcomas the UICC (Union Internationale Contre le Cancer) score; as listed below:

  • GX: Grade cannot be assessed (undetermined grade)
  • G1: Well-differentiated (low grade)
  • G2: Moderately differentiated (intermediate grade)
  • G3: Poorly differentiated (high grade)
  • G4: Undifferentiated (high grade)

- Please include the subtypes: corrected accordingly at line 89-90

- Figure 1: corrected accordingly

- Figure 2: corrected accordingly

- Figure 3: exclusion is clarified at line 157-158

- Figure 4 and 5: corrected accordingly with caption at the side of the graphics

- Lines 194-199 were rewritten accordingly

- Lines 200-202 corrected accordingly

We are grateful for the time and effort you have dedicated to reviewing our submission.

Once again, thank you for your kind words and support. We look forward to the next steps in the publication process.

Best regards,

S.Consalvo

Reviewer 3 Report

Comments and Suggestions for Authors

Chondrosarcomas are the second most common oncological bone disease in humans, primarily affecting men and adult populations. An essential factor in the growth and spread of tumors is systemic inflammation. An indicator of inflammation, C-reactive protein (CRP) is frequently raised in a number of human malignancies. It has been demonstrated that in certain cases of bone and soft tissue sarcomas, a higher level of CRP prior to surgery is associated with a lower overall survival rate. It should be highlighted that less than 30 research have found a correlation between CRP and the prognosis of sarcomas, despite the fact that PubMed lists over 3,000 distinct publications on CRP in malignancies. It follows that the relevance of this piece is clear. The study's objective, according to the authors, was to determine whether the CRP level prior to surgery had any predictive value for the length of survival and whether there was any potential relationship between the CRP level and the histological type of chondrosarcoma. A retrospective analysis based on patient treatment data from a single hospital, careful patient selection for this study, thorough pathohistological research, and adequate statistical analysis should be the work's strong points. The article is written in a conventional style, and the summary gives a thorough overview of the study's primary findings and its main thrust. The research was carried out in compliance with the licenses issued by the facility where the patients were receiving care. The authors go into great detail about the patient selection plan in the materials and methods section. The key points are the CRP parameters, the process for evaluating the pathohistological examination of surgical material, the criteria for determining survival, and statistical analysis techniques, such as Pearson correlation analysis. In this context, the question of why correlations were found using Pearson rather than Spearman's methodology emerges. Using Spearman's technique instead of Prison's makes more sense based on the variation in the CRP level. The study's findings can be replicated if data from patients with chondrosarcomas and the level of CRP are available, although coincidences in the location of metastases and T Status are rare. Both tables and graphs, which make it easy to assess the study's findings, are used to present the work effectively. The authors verified their hypothesis regarding a potential correlation between CRP levels and histology findings during the retrospective analysis. The authors draw the conclusion that patients with CRP levels above 0.5 mg/dl have a high risk of poor survival and that it is important to closely monitor the level of CRP prior to surgery.

Though, as I mentioned before, there are relatively few such publications, the work's weak point can be attributable to a small fraction of citations with a period of less than 5 years from the date of publication of articles. There are no knowledge gaps in the research field, and the references provided in the paper are adequate. A dot is positioned before the footnote on lines 37 and 248, and there are two dots at the end of the sentence on line 281, making it possible that the abbreviation on line 78 cannot be understood because it has already been done on lines 19 (in the summary) and 53 (introduction).

Author Response

Pearson and Spearman correlations are both statistical measures used to evaluate the relationship between two variables, but they differ in their underlying assumptions.

The Pearson Correlation analyse a linear relationship. It measures the strength and direction of the linear relationship between two continuous variables. On the other hand the Spearman Correlation analyse a monotonic relationship. It assesses the strength and direction of the monotonic relationship between two variables, which means the variables tend to change together but not necessarily at a constant rate. After consulting our collegues at our statistical Department we chose the Pearson Correlation  because it is more sensitive to outliers, which can significantly affect the correlation coefficient as it is based on the mean values. Outliers can skew the results, leading to misleading conclusions about the strength of the relationship.

We corrected the dots at line 37,248 and 281.

The abbreviation at line 78 and 53 were also corrected

We are grateful for the time and effort you have dedicated to reviewing our submission.

We look forward to the next steps in the publication process.

Best regards,

S.Consalvo